# Pathogenesis of Choledochal Cyst: Insights from Genomics and Transcriptomics

**DOI:** 10.3390/genes13061030

**Published:** 2022-06-08

**Authors:** Yongqin Ye, Vincent Chi Hang Lui, Paul Kwong Hang Tam

**Affiliations:** 1Faculty of Medicine, Macau University of Science and Technology, Macau, China; 2202853ghp30001@student.must.edu.mo; 2Department of Surgery, School of Clinical Medicine, University of Hong Kong, Hong Kong, China; vchlui@hku.hk

**Keywords:** choledochal cyst, pathogenesis, genomics, transcriptomics

## Abstract

Choledochal cysts (CC) is characterized by extra- and/or intra-hepatic b\ile duct dilations. There are two main theories, “pancreaticobiliary maljunction” and “congenital stenosis of bile ducts” proposed for the pathogenesis of CC. Although family cases or CC associated with other anomalies have been reported, the molecular pathogenesis of CC is still poorly understood. Recent advances in transcriptomics and genomics analysis platforms have unveiled key expression signatures/genes/signaling pathways in the pathogenesis of human diseases including CC. This review summarizes insights from genomics and transcriptomics studies into the pathogenesis of CC, with the aim to improve (i) our understanding of its underlying complex pathomechanisms, and (ii) clinical management of different subtypes of CC, in particular their associated hepatic fibrotic change and their risk of malignancy transformation.

## 1. Introduction

Choledochal cysts (CC), also known as congenital biliary dilation, presents as the dilation of extrahepatic bile ducts. It has a worldwide incidence of about 1:100,000 to 1:150,000 but the incidence can be as high as 1 in 1000 in Asians. In addition, the disease incidence shows a female predominance, with a female to male ratio of 3~4:1. CC is often seen in infants, and can be diagnosed incidentally or when it causes biliary pancreatitis with abdominal pain, jaundice, and even liver enlargement. According to Todani’s classification [1], CC can be anatomically classified into 5 major types (type I–V). Type I, a dilation of extrahepatic common biliary duct (CBD), and type IV, a dilation of both intrahepatic bile duct and extrahepatic CBD, are the two most common types, accounting for about 50–80% and 15–35% of all the CC cases, respectively. Type II, a diverticulum of CBD, and Type III, a dilation of the duodenal portion of the CBD that bulges into duodenal lumen, are rarely seen in infants. Type V, also called Caroli disease, is characterized by multiple dilations of the intrahepatic bile ducts, and the disease is always accompanied with polycystic kidney disease (PKD) or congenital liver fibrosis (Caroli syndrome).

The etiology of CC remains unclear. There are two main theories, “pancreaticobiliary maljunction” and “congenital stenosis of bile ducts”, proposed for the pathogenesis of CC [2,3,4,5]. Pancreaticobiliary maljunction is a congenital malformation in which the pancreatic and bile ducts join outside the duodenal wall, forming an abnormally long common channel. In patients with pancreaticobiliary maljunction, the sphincter of Oddi fails to regulate the function of the pancreaticobiliary junction, resulting in a two-way regurgitation and a mixing of pancreatic juice and bile, which activates the pancreatic enzymes in the pancreatic juice, leading to bile duct damages and inflammation, which subsequently result in a dilation of the bile duct as seen in CC. However, pancreaticobiliary maljunction can only be identified in about 50–80% of patients with CC [6]. Congenital stenosis of bile ducts is characterized by a reduced number of ganglions and neurons of the CBD, which causes a dysregulated contraction and increased intralumenal pressure in the proximal bile duct, leading to a cystic distention of the proximal CBD in CC [7,8,9,10].

There is evidence suggesting a genetic contribution for CC. For example, some CC cases were reported in families [11,12,13], some CC cases were found to be associated with familial adenomatous polyposis [14,15], and Caroli disease was often associated with autosomal recessive polycystic kidney disease (ARPKD) or autosomal dominant polycystic kidney disease (ADPKD) [16,17,18,19,20], and the disease incidence varies between genders and races. The following sections summarize recent insights into the pathogenesis of CC from genomics and transcriptomic studies.

### 1.1. Chromosomal Anomalies

Chromosomal anomalies have been identified in CC (Table 1). Chromosomal duplications such as 17q12 are seen in type Ia CC [21]. 17q12 chromosomal region contains the *HNF1B* gene, and *HNF1B* mutations or 17q12 microdeletions are usually seen in patients with nephritic cysts, diabetes syndrome, and neurologic and psychiatric symptoms [22,23,24,25,26,27]. *HNF1B* encodes a transcription factor called hepatic nuclear factor-1 β (HNF1B), which is involved in bile duct organogenesis [28], and its deletion has been shown to induce jaundice, and anomalies of gall bladder [29]. Kettunen et al. evaluated 14 patients with *HNF1B* mutations and found 6 patients had CC, which exceeds the overall incidence of CC in the general population, indicating that mutations *HNF1B* in 17q12 may play a role in the pathogenesis of CC [30]. Later, Kotalova et al. reported that a two-week-old boy carrying a 17q12 duplication spanning 1698 kb that contains the *HNF1B* gene also suffered from CC, which is similar to the phenotype as seen in patients with a 17q12 microdeletion. The 17q12 duplication led to an overdosage of *HNF1B* and was found both in the young boy and his mother, indicating that the autosomal dominant inherence may exist, although his mother was completely within normal range [21].

De novo 2p15p16.1 microdeletion encompassing the *CCT4*, *COMMD1*, *B3GNT2*, and *EHBP1* genes has been shown to associate with multiple congenital anomalies including renal anomalies, intractable seizures, and a choledochal cyst [31,32,33,34,35,36,37]. Among these genes, *EHBP1*, the gene-encoding and actin-binding protein for cytophagy, was the only gene deleted in a type II CC patient [34]. Chromosome arm 5q deletion has been widely reported in familial adenomatous polyposis (FAP), which is due to mutations of the *APC* gene [38,39]. Deletion of chromosome 5q was once reported in a type V CC (Caroli disease) patient with FAP [14]. A cytogenetic study of a patient with a familial isolated Caroli disease revealed an unbalanced translocation between chromosome 3 and chromosome 8 with t(3;8) [40]. Genes located in 3p23 and 8q13 were rearranged, indicating that the distal loss and/or gain of 8q may contribute to the pathogenesis of Caroli disease. Genomic gains of 8q can promote ribosome biogenesis activity, which promotes hepatocellular carcinoma development [41,42]. CC has a higher risk of malignancy transformation [43,44,45], although a link between chromosomal translocation t(3;8) leading to a gain of 8q13-qter and the risk of malignancy transformation in CC has yet to be established.

### 1.2. Transcriptomics Analysis

There have been few microarray or RNA sequencing studies on the etiology of CC. Lv et al. performed transcriptome sequencing analysis on subtypes of Type I CC including cystic CC (Ia) and fusiform CC (Ic), and identified 6463 differentially expressed genes (DEGs) implicated in the biological processes of epithelial cell differentiation or extracellular matrix between these two CC subtypes [46]. Pathways related to metabolism and hormone regulation were enriched in cystic forms, while pathways related to immune response were enriched in fusiform CC [47,48]. An enrichment of immune-related genes is in line with a common clinical manifestation of immune-related complications including cholangitis and pancreatitis in fusiform CC [47,48]. Weighted gene co-expression network analysis (WGCNA) enables identification of gene modules and key genes that were correlated and relevant to clinical traits [49,50]. Using WGCNA, 12 co-expression modules were constructed, and the blue module, comprising the second most genes, was identified to be strongly correlated with fusiform CC [46], indicating that genes within this module could be potential markers for subtypes of CC. To further investigate key candidate genes in this module, a protein–protein interaction (PPI) was performed and found that the blue module contained key genes enriched in the Wnt signaling pathway, and activation of the Wnt pathway was associated with cholangio-carcinogenesis [51,52]. CC has a higher risk of malignancy transformation; whether an activation of Wnt pathway is involved in malignancy transformation of CC requires further investigation. *ERBB2* and *WNT11* were the two hub genes in this same module that distinguished fusiform CC and cystic CC. *ERBB2* mutations/amplifications were implicated in extrahepatic cholangiocarcinoma [53,54,55]. All of these indicated that the CC subtypes share some common transcriptomic signatures, but each one also displays its own unique transcriptomic signatures. Therefore, elucidation of their common and unique transcriptomic signatures for each CC subtype will not only provide insights into the molecular pathogenesis of each subtype but also help in clinical management of the disease. For example, if the subtype’s unique transcriptomic signature suggests an elevated risk of cancer development, a more frequent follow-up and closer monitoring of any pre-malignant changes in the liver of post-surgical patients will help to identify patients for early cancer treatment.

### 1.3. Genetic Variants

A trio-based whole exome-sequencing of 31 CC trios was performed to identify de novo variants [56]. A total of 27 non-synonymous de novo variants were identified, 21 of which were damaging de novo variants, including 4 protein truncating mutations and 17 missense mutations. The constraint scores on average are statistically higher than random samples, indicating that these sequence variations are likely to contribute to CC. Six genes (*PXDN*, *RTEL1*, *ANKRD11*, *MAP2K1*, *CYLD*, and *ACAN*) with the de novo variants are involved in human developmental diseases, such as sclerocornea, spondyloepimetaphyseal dysplasia, dyskeratosis congenita, and familial multiple trichoepitheliomata [57,58,59,60,61,62,63,64,65]. Furthermore, four genes (*PIK3CA*, *TLN1 CYLD*, and *MAP2K1*) with the damaging variants are linked to bile duct and liver cancer. De novo variants in significantly mutated regions (SMRs) genes (*PIK3CA*, *C6*, and *PPP2R2B*) were over-represented in CC patients, especially *PIK3CA* with excess mutations implicated in 9 cancer types, indicating that it may play a role in malignancy [66,67,68,69,70]. A total of 12 genes (*DCHS1*, *EPS15*, *DNM1*, *C5orf42*, *POU2F2*, *THBS1*, *POU2F2*, *BYSL*, *C5orf42*, *THBS1*, *BYSL*, *TXLNB*) with de novo damaging variants recurrently identified in different individuals, forming either protein–protein interactions (PPIs) or compound heterozygotes or in homozygous. PPI pair TRIM28 and ZNF382 were overrepresented in CC group compared to controls in PPI analyses, with 3 and 2 patients carrying rare damaging variants on these two genes, respectively. TRIM28, known for regulation of the endoderm differentiation and involvement in malignancy, may play a role in CC [71,72,73,74]. Though the whole exome sequencing study was limited by a small sample size, CC is regarded as a multigenetic disease with mutations in more than one gene with genetic heterogeneity [56].

### 1.4. Genetics in Type V CC

Type V CC, also called Caroli disease, is a congenital disorder and rare genetic condition characterized by multiple dilations of intrahepatic bile ducts, with an incidence of around 1 in 1,000,000 [75]. When it is associated with congenital hepatic fibrosis, it is referred to as Caroli syndrome, which has a higher incidence of 1 in 100,000 [76]. The disease has been shown to display either autosomal recessive (AR) or autosomal dominant (AD) mode of inheritance [11,12,13,77,78,79,80], usually accompanied with polycystic kidney disease. Since the simple type (Caroli disease) is rare compared to the periportal fibrosis type (Caroli’s syndrome), knowledge of its pathogenesis is very limited. Ductal plate malformation (DPM) may play a role in the pathogenesis of Caroli disease, which leads to persistent embryonic bile ducts after birth [81,82]. Beyond DPM, Takahashi et al. identified that activation of Notch signaling involving Notch1 and Hes1 may also participate in the progression of biliary cystogenesis in Caroli disease [83].

*PKHD1* encodes a protein called fibrocystin that is involved in kidney organogenesis, and mutation of *PKHD1* gene is responsible for cyst formation in kidney. *PKHD1* gene mutations were identified in Caroli disease associated with autosomal recessive polycystic kidney disease (ARPKD) [16,20,79,84]. Fibrocystin was localized in the cilia of bile duct cells, and a loss-of-function mutation of *PKHD1* causes cilia dysfunction with shortening and deformation of cilia, leading to an abnormal Wnt signaling and ductal plate malformation [20]. Three missense variants (Tyr1136Cys, Arg1369Cys, and Thr2869Lys) of *PKHD1* were identified in a patient with Caroli syndrome, these mutations led to a replacement of the polar amino acid by the less polar amino acid in the extracellular domains of fibrocystin affecting fibrocystin deposition/function in the cilia [85]. Another missense variant (Lys626Arg) of *PKHD1* was seen in Caroli syndrome as a paternal inheritance [86]. Later, the same paternally inherited *PKHD1* missense mutation was seen in Caroli syndrome associated with type II Abernethy malformation (a rare portal venous system anomaly) [87].

Mutations in polycystin-1 encoding gene *PKD1* or polycystin-2 encoding gene *PKD2* were commonly seen in cases of Caroli disease associated with autosomal dominant polycystic kidney disease (ADPKD). Caroli disease has also been identified in families with a young sister, older brother, and father, which suggests that the disease could be inherited in an autosomal dominant mode, although autosomal recessive mode of inheritance was still the main mode of inheritance [88]. More familial Caroli cases in autosomal dominant inheritance mode have been reported since [17,19,89,90,91,92]. Polycystin-1 and polycystin-2 are involved in cell–cell or cell–matrix interactions as part of a multiprotein membrane-spanning complex [93,94]. A mutation of the exon 46 (12801de128) of the *PKD1* gene was associated with Caroli disease and PKD1 phenotype in two family members [17]. Follicle-stimulating hormone (FSH) may promote renal and liver cystic growth via the cAMP/ERK signaling pathway [90].

Pathogenic variants in the *WDR19*/*NPHP13* gene were first reported in Caroli disease associated with nephronophthisis-related ciliopathies (NPHP-RC), an autosomal recessive cystic kidney disorder [95]. The same mutation was discovered in a subsequent exome sequencing study on 48 Koreans with NPHP in that 6 of them suffered from Caroli disease [18]. *GLIS2*/*NPHP7* recessive missense mutation was also found in NPHP-RC [95]. Homozygous mutation of NPHP3 may cause ciliopathy malformation complex (Caroli disease, bilateral cystic renal dysplasia, and postaxial polydactyly), which follows an autosomal recessive inheritance mode with a 25% recurrence risk in every pregnancy [96] (Table 2).

### 1.5. Animal Model

Katsuyama et al. first established a novel PKD model (named as PCK rat) with multiple cysts in liver and kidney, by sib mating of a female rat derived from Crj:CD (SD) rats [97]. Unlike other polycystic kidney animal models, PCK rat developed a complete polycystic malformation in liver and kidney. Sanzen et al. found that the cysts in the PCK rats were cystic dilations of the intrahepatic bile ducts, indicating that PCK rats are a useful animal model for studying Caroli disease and congenital hepatic fibrosis (CHF) [98]. Using PCK rats, investigators observed ductal plate malformation, decreased laminin, type IV collagen production, over-secretion of plasminogen and tissue-type plasminogen (tPA) from bile duct cells, associated with biliary dysgenesis and progressive cystic dilation in these rats [82,99]. Furthermore, activation of MEK5-ERK5 cascade has also been identified in PCK rats [100]. MEK5-ERK5 signaling cascade is involved in laminin and type IV collagen deposition at the basal membrane of bile duct cells; hence, dysregulation of MEK5-ERK5 signaling could perturb cell–cell interactions of bile duct cells and liver parenchyma, leading to biliary dysgenesis and cystogenesis in PCK rats [101].

### 1.6. Malignancy Transformation

Choledochal cysts are associated with an increased risk of cancer, particularly cholangiocarcinoma (70.4%) but also pancreatic and gallbladder cancers (23.5%), and especially those with type I and IV cysts [45]. Although the incidence of malignancy was pretty low (0.4%) in patients below 18 years old, it increased with each decade, and the incidence in adult patients could be as high as 11.7% [102], and it can be up to 30–40% of those over 50 years [103,104]. In general, the overall incidence of malignancy transformation was 7.5% in a review of 5780 patients with CC, which was 1000–2000 times higher than that in the general population [45,105]. Median age for cholangiocarcinoma was around 42 years old, 20 years younger than those without CC [45,105]. Schwab et al. retrospectively reviewed CC patients who underwent surgical resection and found that 1/48 developed cholangiocarcinoma [106]. *TP53* and *RBM10* mutations have been identified in those patients, which showed an 8.9% and 1.1% frequency for cholangiocarcinoma from CC, respectively [107,108,109]. *KRAS* amplification was also discovered. *KRAS* mutation was first seen in a gallbladder adenoma of a patient with FAP, suggesting that *KRAS* mutation may induce malignancy transformation [110]. Immunohistochemical studies of bile duct epithelium tissues from CC patients showed elevated expressions of Ki67, KRAS, and p53 in both bile duct epithelium and gallbladder epithelium, suggesting that KRAS may be involved in cancer development of these patients [111]. Expression of inducible nitric oxide synthase (iNOS) also increased significantly in CC, which may contribute to biliary mucosal hyperplasia, inflammation, and malignant transformation of the bile duct mucosa [112].

### 1.7. Potential Future Research Directions

Recent genomics and transcriptomics studies have yielded important insights into the complex pathogenesis of CC, but our knowledge of genetic heterogeneity and the mechanisms underlying the causation/progression and malignancy transformation of CC is still in its infancy. To demonstrate the causal links between the genomic and transcriptomic signatures and CC will be a daunting task. Application of CRISPR/Cas9-based genome editing in patients’ liver tissue-derived and human iPSC-derived liver organoids could help in the functional evaluation of genomic variants and dysregulation of genes/pathways in the pathogenesis of CC.

Due to the genetic heterogeneity and its underlying complex pathomechanisms, it is likely that CC is caused by the joint contribution of a number of independently acting or interacting polymorphic genes, each displaying different effect sizes that act simultaneously or interactively in the causation and progression of the disease. Therefore, to unravel the pathogenesis of CC, a combined analysis of the huge database of genomic variants and transcriptomic architectures, together with a genotype–phenotype correlation analysis of their clinical manifestations, associated anomalies, and malignant transformation are essential, and such analysis is too large or complex to be efficiently dealt with by traditional data-processing application software.

Artificial intelligence (AI) solves complex problems by mimicking human brains. Using supervised, semi-supervised, and unsupervised learning, as well as evolutional manners, AI identifies human genetic patterns and disorders from genetic data [113,114]. Deep learning, a specific AI algorithm, has been successfully applied for variant calling and classification, phenotype-to-genotype and genotype-to-phenotype mappings and classifications, predicting transcriptomic profiles and clinical diagnosis, and improving precision medicine for complex diseases by the analysis of complicated and large-scale genomic databases [113,114,115]. The application of AI algorithms in the analysis of massive data of genomics/transcriptomics/clinical data generated from the CC studies could identify genetic variants/transcriptomic profiles associated with different subtypes of CC, improving our knowledge of their underlying disease mechanisms and their associated risk of malignancy.

## 2. Conclusions

After decades of research on choledochal cysts, its causes remain unclear. It is generally believed that mutations in multiple genes with genetic heterogeneity drives this complex disease. Recent advances in transcriptomics and genomics analysis have unveiled key expression signatures/genes/signaling pathways in the pathogenesis of subtypes of CC and their relevance in the malignant transformation of CC. Elucidation of the common and unique genomics and transcriptomic signatures for each CC subtype not only provides insights into their complex molecular pathogenesis but also improves clinical management of the disease.

## 3. Search Strategy and Selection Criteria

Data for this review were collected from PubMed and Web of Science databases using the Mesh terms “Choledochal Cyst/genetics’’ or keywords “choledochal cyst” or “common biliary dilation” and “genetics”. All article types published between 1992 and 2022 were included. Each relevant text was thoroughly reviewed and examined, including relevant papers in the references list. Articles were excluded from review if the paper was not in English or the full-text was unobtainable (Figure 1).

## Figures and Tables

**Figure 1 genes-13-01030-f001:**
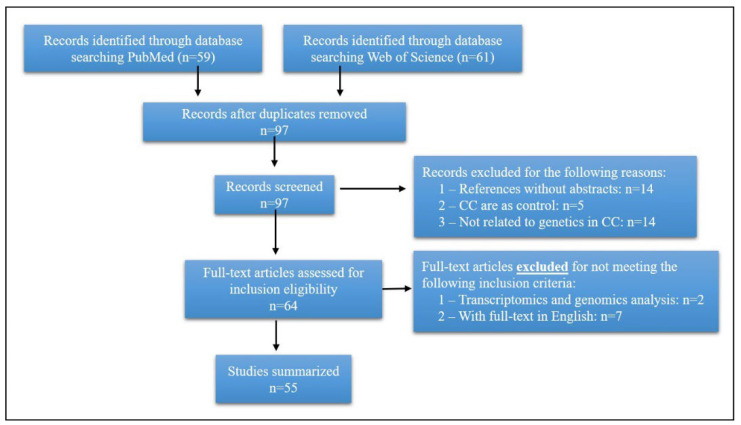
Flow-chart showing the search strategy and inclusion/exclusion criteria used in the review.

**Table 1 genes-13-01030-t001:** Chromosomal alterations reported in choledochal cyst patients.

Chromosomal Anomaly	Localization	Phenotype Description	Type	Candidate Gene	References
Duplication	17q12	Diabetes, renal disorders, structural brain anomalies, etc.	Ia	*HNF1B*	[21]
Deletion	Microdeletion 17q12	Diabetes, renal cysts, neurological and psychological anomalies, biliary or hepatic clinical phenotype	I, IV	*HNF1B*	[30]
Deletion	2p15p16.1	facial features, developmental delay Congenital microcephalyand intractable myoclonic epilepsy	II	*EHBP1*	[34]
Deletion	5q	Familial adenomatous polyposis	V	-	[14]
Translocation	t(3;8)(p23;q13)	Abdominal pain, jaundice	V	*APC*	[40]

**Table 2 genes-13-01030-t002:** Gene mutations relevant to CC associated with other anomalies.

Mutation	Disease	References
*PKHD1*	CD + ARPKD	[16,20,79,84]
*PKHD1*	CS + type II Abernethy malformation	[87]
*PKD1, PKD2*	CD + ARPKD	[17,19,89,90,91,92]
*WDR19/NPHP13*	CD + NPHP-RC	[95]
*GLIS2/NPHP7*	CD + NPHP-RC	[95]
*WDR19*	CD + NPHP 13	[18]
*NPHP3*	Ciliopathy malformation complex	[96]

ARPKD: autosomal recessive polycystic kidney disease; CS: Caroli syndrome; CD: Caroli disease; NPHP-RC: nephronophthisis-related ciliopathies; NPHP 13: nephronophthisis 13; Ciliopathy malformation complex: Caroli disease, bilateral cystic renal dysplasia, and postaxial polydactyly.

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
