# Peer review of "Pathogenesis of Choledochal Cyst: Insights from Genomics and Transcriptomics"

_genes, 2022, doi:10.3390/genes13061030_

Round 1
Reviewer 1 Report
This review summarized recent advances in pathogenesis of Choledochal Cysts (CC), especially from the perspective of genomic and transcriptomic analysis. Also, animal model and risk of cancer were included. This is a good work trying to give us a complete understanding of CC. But there are still some questions need to be addressed:
1. This work only summarized previous work, the discussion part is missing. perspectives and directions for future study are more valuable for this field.
2. In the part of malignancy transformation, is there any difference between children and adults?
3. In the part of type V CC, pathogenesis of isolated or simple form of caroli disease should also be discussed.
4. Reorganize this sentence "Mutations of WDR19 were also identified in Caroli disease asso-175 ciated with NPHP 13 in a Korean population [18]." as it diescribes the same gene mentioned in line 172.
5. Some other mistakses, as in line74, "type V" better than "type 5", in line146, the gene name should be PKHD1.
Author Response
Reviewer 1
Comment 1: This work only summarized previous work, the discussion part is missing. perspectives and directions for future study are more valuable for this field.
Response: A new paragraph on the potential future research directions on CC was added to the revised manuscript (ln 229-ln 258).
Comment 2: In the part of malignancy transformation, is there any difference between children and adults?
Response: This reviewer right points out that, and indeed there are differences between adults and children in terms of incidence. The incidence of malignancy before the age of 18 was 0.42% versus 11.4% in adults. In general, cholangiocarcinoma was more found than gallbladder cancer (70.4% vs. 23.5%), however, there are no evidence which specific type of cancer are more prone to develop in adult and children group. We have included the above statistics of malignancy transformation in paediatric and adult patient groups with new references in the revised manuscript (ln 209-ln 216).
Comment 3: In the part of type V CC, pathogenesis of isolated or simple form of Caroli disease should also be discussed.
Response: We have added a paragraph on the potential pathogenesis of isolated form of Caroli disease with new references in the revised manuscript (ln 147-ln 152).
Comment 4: Reorganize this sentence "Mutations of WDR19 were also identified in Caroli disease asso-175 associated with NPHP 13 in a Korean population [18]." as it describes the same gene mentioned in line 172.
Response: The sentences have been rewritten (ln 181-ln 183).
Comment 5: Some other mistakes, as in line74, "type V" better than "type 5", in line 146, the gene name should be PKHD1.
Response: Sorry for our inaccuracy. The mistakes have been corrected in the revised manuscript.
Reviewer 2 Report
The authors submitted a review manuscript entailed "Pathogenesis of Choledochal Cyst: Insights from Genomics and Transcriptomics." The manuscript is overall qualified. However, there are some concerns about this article. 1 Even a narrative review, it is necessary to present the flow chart of literature search methods, including inclusion/exclusion criteria. 2. It is not clear what this study's primary /secondary outcome is. 3. The number of references is enough for a full review article; however, these are not updated. 4. Utilizing artificial intelligence in this field of study should be discussed.
Author Response
Comment 1: Even a narrative review, it is necessary to present the flow chart of literature search methods, including inclusion/exclusion criteria.
Response: A flow chart of the search methods (Fig.1) has been added to the revised manuscript.
Comment 2: It is not clear what this study's primary / secondary outcome is.
Response: Primary outcome for this study is to summarize the potential genetic mechanisms to improve our understanding of its underlying complex pathomechanisms. Secondary outcome for this study is to provide current advances in this field to improve clinical management different subtypes of CC, in particular their associated hepatic fi-brotic change and their risk of malignancy transformation. We have modified the abstract to clarify the aims of this review in the revised manuscript (ln 14-ln 17).
Comment 3: The number of references is enough for a full review article; however, these are not updated.
Response: We added some new paragraphs with additional more up-to-date citations in the revised manuscript.
Comment 4: Utilizing artificial intelligence in this field of study should be discussed.
Response: We address the potential usage of AI in the future study for CC (ln 239-ln 259).
Round 2
Reviewer 2 Report
It is well revised according to the reviewers' comments. No further comments are available.